# Using Multilevel Regression and Poststratification to Estimate Physical Activity Levels from Health Surveys

**DOI:** 10.3390/ijerph18147477

**Published:** 2021-07-13

**Authors:** Marina Christofoletti, Tânia R. B. Benedetti, Felipe G. Mendes, Humberto M. Carvalho

**Affiliations:** Department of Physical Education, School of Sports, Federal University of Santa Catarina, Florianópolis 88040-900, SC, Brazil; marinachriss@outlook.com (M.C.); tania.benedetti@ufsc.br (T.R.B.B.); felipe_goedert@hotmail.com (F.G.M.)

**Keywords:** survey methods, Bayesian analysis, public health surveillance, selection bias, statistical models

## Abstract

Background: Large-scale health surveys often consider sociodemographic characteristics and several health indicators influencing physical activity that often vary across subpopulations. Data in a survey for some small subpopulations are often not representative of the larger population. Objective: We developed a multilevel regression and poststratification (MRP) model to estimate leisure-time physical activity across Brazilian state capitals and evaluated whether the MRP outperforms single-level regression estimates based on the Brazilian cross-sectional national survey VIGITEL (2018). Methods: We used various approaches to compare the MRP and single-level model (complete-pooling) estimates, including cross-validation with various subsample proportions tested. Results: MRP consistently had predictions closer to the estimation target than single-level regression estimations. The mean absolute errors were smaller for the MRP estimates than single-level regression estimates with smaller sample sizes. MRP presented substantially smaller uncertainty estimates compared to single-level regression estimates. Overall, the MRP was superior to single-level regression estimates, particularly with smaller sample sizes, yielding smaller errors and more accurate estimates. Conclusion: The MRP is a promising strategy to predict subpopulations’ physical activity indicators from large surveys. The observations present in this study highlight the need for further research, which could, potentially, incorporate more information in the models to better interpret interactions and types of activities across target populations.

## 1. Introduction

Leisure-time physical activity has beneficial health effects [1]. It is an essential asset to encourage physical activity in population-based programs [2]. National Health surveys are an indispensable resource for developing health promotion programs, including the promotion of physical activity practices. Hence, data about health-related behavior considering physical activity and life quality are valuable [3].

Sociodemographic characteristics and environmental and contextual variation are essential determinants of physical activity [4]. Furthermore, differences in sociodemographic factors on leisure-time physical activity may promote and successfully implement healthy practices and lifestyles. Sociodemographic characteristics have been necessary to understand the current health scenario [5].

National health surveys often include information about sociodemographic characteristics and several health indicators that often vary across communities, regions, or states. The common problem is that samples of respondents in a survey for some units at the community, region, or state level are too small and often not representative of the larger population. This represents a necessary methodological hurdle for research on health and physical activity when making valid inferences from a collected survey sample to the larger (underlying) population or subpopulations [6].

To deal with this limitation in extensive surveys, researchers have used disaggregation, i.e., the no-pooling approach, where the outcome information is used solely from the survey respondents’ subpopulations. The no-pooling approach assumes that each subpopulation provides no information about any other subpopulation [7]. Disaggregation from even extensive surveys often produces small samples and noisy outcome estimates [8].

On the other hand, inferences from health-related survey data, particularly considering physical activity outcomes, have generally used single-level regression models to combine relevant information from individual and contextual characteristics [9,10,11,12,13,14,15,16]. Physical activity research often explores associations of physical activity and health indicators with individuals’ characteristics, such as gender, age, marital status, ethnicity, work status or education level, and geographical characteristics, such as community, city, region, or country levels. This often presents a cross-classified and/or hierarchical data structure. However, a single-level regression model, i.e., a complete-pooling approach, assumes that the subpopulations are invariant, the same as estimating a standard parameter for all subpopulations [7]. Furthermore, with imbalanced sampling common in surveys, when some individuals, locations, or times are sampled more than others, over-sampled clusters likely dominate the inference [7].

Alternatively, multilevel regression and poststratification (MRP) has become a standard modeling approach to estimate subnational or subpopulations outcomes in large-scale surveys [6]. MRP was developed [17,18] and has been mainly used in the political sciences [6,8,19,20,21]. In this context, MRP has been noted to outperform disaggregated empirical means [19,21]. Recently, MRP has been applied to health science data [22,23,24,25,26]. The approach initially uses multilevel regression to model individual outcomes of interest as a function of individuals and/or contextual and geographical predictors to estimate a target subpopulation [27]. Lastly, the outcome estimates for each individual–contextual subgroup are weighted by each subgroup’s proportions in the actual population to derive an overall population-level estimate [19,21,23]. The key to the superiority of MRP is in the multilevel model used that allows for more efficient use of the data. The multilevel model allows for partial pooling by incorporating group-level effects (also referred to as random effects). These may be understood as a weighted average between the total sample estimate and a group or unit estimate, where the specific weights are based on the entire variation of the sample and the group or unit variation [27]. In particular, the partial pooling will be more substantial for smaller units with fewer observations [27]. Moreover, it has been highlighted that the careful inclusion of contextual or geographical variables (higher hierarchical variables) can improve the prediction precision of MRP [21,23]. In the context of physical activity surveys, the inclusion of simple demographic information may suffice for MRP, as noted in other fields [19,21], but the addition of demographic information and increase in model complexities to improve estimations merit further study.

Brazil offers a particular case study. It is one of the world’s most populated countries, with extensive demographic, socioeconomic, and cultural contrasts. Additionally, there have been several national health surveys implemented to support the country’s health surveillance system. The surveys include at least three primary large-scale national health surveys: the National Health Survey (PNS) [28], the National Adolescent School-based Health Survey (PeNSE) [29], and the “Surveillance of Risk Factors and Protection Against Chronic Diseases by Telephone Inquiry” (VIGITEL) [30]. These surveys have provided an essential resource of information to support the development of health promotion programs at national, state, and community levels, including the promotion of physical activity practices. Nevertheless, interpretations have been mostly based on single-level aggregated models [31,32,33,34].

This study developed an MRP model to estimate the proportion of individuals with at least 150 min per week of leisure-time physical activity across Brazilian state capitals, and it considers age groups and gender as demographic characteristics. Hence, we adopted a secondary data analysis from the annual national survey VIGITEL. We compared competing model estimates of leisure-time physical activity across Brazilian state capitals to evaluate whether the MRP approach outperforms single-level regression estimates. Lastly, we estimated and interpreted the proportion of individuals with at least 150 min per week of leisure-time physical activity estimated using MRP across subpopulations: female and male individuals in each Brazilian state capital and age group.

## 2. Materials and Methods

### 2.1. Data

We used the responses from the annual national survey VIGITEL conducted in 2018 in all 27 capitals of the Brazilian states (available at http://svs.aids.gov.br/download/Vigitel/, accessed on 3 February 2021), and the demographic data from the Brazilian census of 2010 of the Brazilian Geography and Statistics Institute (IBGE) (available at https://www.ibge.gov.br/estatisticas/sociais/populacao/9662-censo-demografico-2010.html?edicao=9673&t=downloads, accessed on 3 February 2021). The VIGITEL annual sampling included at least 2000 interviewees in each state capital and assumed that that the outcomes could be estimated with a 95% confidence interval and a 3% maximum error [35]. The survey used raking to establish weighting factors to compensate for bias of non-universal fixed-line telephone coverage, adjusted to the adult Brazilian population based on the weight of each individual of the sample [35]. Hence, the survey is assumed to be a relatively representative and balanced sample of the state capitals based on the Brazilian population [35]. In this study, the VIGITEL survey sample included responders who were at least 20 years old and offered an outcome response (physical activity practice in leisure time), which totaled 47,121 individuals. The outcome variable, the physical activity level in leisure time, was categorized into inactive (<150 min/week) and active (≥150 min/week). We considered gender (two levels: male and female), age group (six levels: 20 to 29, 30 to 39, 40 to 49, 50 to 59, 60 to 69, and more than 70 years old), from all 27 states capitals in Brazil. The VIGITEL survey was approved by the National Committee of Ethics in Research with Human Beings of the Ministry of Health [30,35].

### 2.2. Data Analysis

#### 2.2.1. Multilevel and Poststratification

The approach initially uses multilevel regression to model individuals’ responses as a function of both demographic (gender and age group) and geographic (state capital) predictors, partially pooling the outcome variable (amount of physical activity per week). We use a multilevel logistic regression model. We labeled the survey response *y_i_* as 0 for physically inactive individuals (physical activity below 150 min/week), and as 1 for physically active individuals (physical activity above 150 min/week). Each individual’s outcome was estimated as a function of the individual’s characteristics, i.e., age group, gender, and state capital (for individual *i*, with indexes *j*, *k*, and *l* for gender, age group, and state capital, respectively). We considered age as a population-level effect (also referred to as fixed effect), given the difficulty of estimating the between-group variation when the number of groups is small [27]:(1)$$Pr\;\left(y_{i}=1\right)\sim \;logit^{-1}\left(\beta_{j\left[i\right]}^{gender}+\alpha_{k\left[i\right]}^{age\;group}+\alpha_{l\left[i\right]}^{state\;capital}\right)$$
where gender as two levels (*j* = 1, 2), age group as 7 levels (*k* = 1, …, 7), and state capital as 27 levels (*l* = 1, …, 27).

The poststratification allows us to estimate the physical activity level attained per week for any set of individual demographic and geographic values, cell *c*, based on the multilevel regression estimates. Hence, the model estimates for each individual’s demographic and geographic group are weighted by each group’s percentages in the actual population, obtained from the most recent Brazilian census in 2010, to produce the MRP preference estimate. The poststratification table comprises 2 gender levels, 7 age-group levels, and 27 state capital levels, encompassing 378 cells (2 × 7 × 27), including the sample size and respective proportion in each group. The prediction in each cell, *θc*, is weighted by these population frequencies of that cell. For each state capital, the average response is calculated over each cell *c* in state capital *s*:(2)$$y_{state\;capital}^{MRP}=\frac{\sum_{c\in s}N_{c}\theta_{c}}{\sum_{c\in s}N_{c}}$$

Hence, it represents our estimations of the proportion of physically active individuals in state *s.*

We used a Bayesian perspective to estimate and interpret the multilevel model fits [36]. The model estimates were regularized using normal prior (0,10) for population-level effects, and exponential prior (1) for group-level effects. We ran three chains in parallel for 500 iterations with 250 warmup iterations. The models were obtained using the “brms” package (Bürkner, 2017), which is written in Stan [37]; this allows us to fit a fully Bayesian model using Hamiltonian Monte Carlo sampling using R [38].

#### 2.2.2. Comparing MRP and Single-Level Regression Models

Single-level aggregated models are often used to analyze health-related outcomes, such as physical activity, cross-sectional observations, and large surveys, despite theoretical and analytical concerns [39,40]. Therefore, we fitted a single-level regression model., i.e., a single-level logistic regression model, to estimate the proportion of physically active individuals. Hence, the single-level model considers gender (*j* = 1, 2), age group (*k* = 1, …, 7), and state capital (*l* = 1, …, 27) as population-level effects:(3)$$Pr\;\left(y_{i}=1\right)\sim \;logit^{-1}\;\left(\beta^{0}+\beta_{j}^{gender}+\beta_{k}^{age\;group}+\beta_{l}^{state\;capital}\right)$$

We used cross-validation based on a split-sample validation approach to compare MRP and single-level aggregated regression estimates’ relative performance [19,21]. The VIGITEL dataset used in this study was randomly split, using half of the sample to define the baseline or “true” proportion of physically active individuals in each state capital. We consider disaggregation means of the baseline subsample as the estimation target [19].

We then used proportions of the remaining subsample to generate estimates of proportions of physically active individuals, employing MRP and single-level regression models. We drew such random samples 300 times for four subsample sizes. The approximate subsample sizes are 23,500 for the baseline subsample, 600 for the 2.5% subsample, 2400 for the 10% subsample, 5900 for the 25% subsample, and 12,000 for the 50% subsample. We intend that the smaller subsample sizes echo typical sizes in small cross-sectional studies and smaller-scaled surveys.

Predictive success, i.e., how close each set of estimates is to the baseline subsample’s target measure, was assessed by calculating the mean absolute error. We follow the notation used in cross-validation studies in MRP in other fields [19,21]. In each run of a simulation *q*, let $y_{q,s}^{baseline}$ be the proportion of physically active individuals in state capital *s* in the baseline data, let $y_{q,s}^{single-levelregression}$ be the single-level regression model estimated proportion in state capital *s* on the sampled data, and let $y_{q,s}^{MRP}$ be the MRP estimate proportion in state capital *s* on the sampled data. Then, for each of the four subsample sizes, we calculated the errors produced by each method in each state capital in each simulation, the most straightforward measure being the absolute difference between the estimates and the baseline measure:(4)$$e_{q,s}^{single-level}=\left|\;y_{q,s}^{single-level}-y_{q,s}^{baseline}\right|,e_{q,s}^{MRP}=\left|\;y_{q,s}^{MRP}-y_{q,s}^{baseline}\right|$$

That forms two matrices of absolute errors, of size 300 (simulations) × 27 state capitals. For state capital *s*, we calculated the mean absolute error for each method across simulations:(5)$${\bar{e}}_{s}^{single-level}=\frac{\sum_{q}e_{q,s}^{single-level}}{300},{\bar{e}}_{s}^{MRP}=\frac{\sum_{q}e_{q,s}^{MRP}}{300}$$

Lastly, we calculated the mean absolute error over both state capitals and simulations, collapsing the means-by-state capital into a simple number for each subsample size and method:(6)$${\bar{e}}^{single-level}=\frac{\sum_{q,s}e_{q,s}^{single-level}}{300\;\cdot \;27},{\bar{e}}^{MRP}=\frac{\sum_{q,s}e_{q,s}^{MRP}}{300\;\cdot \;27}$$

## 3. Results

The VIGITEL sample considered in our analysis comprised 47,121, reflecting those who participated in the survey who were older than 20 years. Distribution by gender, age, and state capitals according to the attainment of recommendations for leisure-time physical activity is summarized in Table 1.

Figure 1 presents a visual overview of the MRP and single-level regression model estimates with the target estimation with different subsample sizes. Moreover, corresponding mean absolute errors of MRP and single-level regression models with different subsample sizes are presented in Figure 2. Note that state capitals are ordered by population size, where Palmas has the lowest population size, and São Paulo has the largest population size. MRP consistently had predictions closer to the estimation target than single-level regression model estimations. Particularly for small subsample sizes, the mean absolute errors were more often smaller for MRP estimates. With larger subsample sizes, mean absolute errors were smaller, closer to zero for both methods than smaller subsample sizes.

Figure 3 displays scatter plots of the estimation target (“true value”) against MRP and the single-level regression model. The single-level regression estimates did not shrink enough to the reference line for perfect correlation with smaller subsample sizes. The MRP estimations were clustered around this line, particularly in smaller subsamples and when compared to single-level regression estimates.

Figure 4 presents an overview of how MRP estimates compare with the VIGITEL single-level regression estimates. MRP presented substantially smaller uncertainty estimates compared to single-level regression estimates.

MRP estimations of the proportions of female and male individuals attaining 150 min per week in leisure-time physical activity across capital states and age groups are displayed in Figure 5. Regardless of state capital or age group, the MRP estimations indicate that males are substantially more physically active than females. By age group, the MRP estimations indicate that younger age groups up to 40 years show a proportion of physically active individuals above 50%. For females, after 30 years, the proportion of physically active individuals is below 40%, decreasing substantially in older age groups.

## 4. Discussion

In this study, based on national health survey data, we compared approaches to estimate leisure-time physical activity across Brazilian state capitals and evaluated whether the MRP approach outperforms single-level regression estimates. We considered a relatively simple MRP model to estimate the proportion of individuals with at least 150 min per week of leisure-time physical activity across Brazilian state capitals and demographic characteristics (age groups and gender). Overall, the results strongly suggest the plausibility of using MRP to estimate health-related outcomes from large-scale surveys. Our simulations showed that MPR estimates outperformed single-level aggregated estimates, mainly when sample sizes were smaller. MRP also showed substantially smaller uncertainty estimates than single-level aggregated modeling (i.e., complete pooling) and disaggregated empirical means (i.e., no pooling). The MRP estimations aggregated by state capital showed, except for one state capital (Brasilia), that the proportion of inactive individuals was substantially larger than active individuals. However, subpopulation estimates by gender across state capitals showed that inactivity was more prevalent in females, and, in several state capitals, there were higher proportions of physically active than inactive male individuals. The MRP estimations showed that only males between 20 and 29 years had a higher proportion of physically active individuals, and males were more active across all age groups than females.

It has been extensively shown in political science research that MRP outperforms survey disaggregation (disaggregated means) [17,19,21,41,42]. We focused on comparing MRP against single-level regression model-based estimations, a common approach when dealing with physical activity outcomes in cross-sectional surveys. Our results are consistent with observations which show that MRP outperforms single-level model estimates [17]. MRP has notably outperformed aggregated estimates in examinations with small sample sizes. Physical activity surveys often comprise sample sizes similar to the range of subsamples considered in this study [10,11,12,13,14,15,16,43,44]. Particularly with small subsample sizes, our estimations showed that MRP had smaller mean absolute errors against a “true value” and less uncertainty in the outcome of interest than single-level regression estimates.

Even with a considerable sample data size and data collection process, the VIGITEL survey has some limitations in its methodology, for instance, the decision to use the only landline, which varies in coverage across the capitals of the North and Northeast [45], and the decision to collect data only from state capital cities. This information did not detract from the VIGITEL survey’s quality and, notably, its yearly data collection but limits its representativeness of the population. Brazil has another national survey, which happens in all cities across the country, giving more national representation: the PNS survey. However, it happens less frequently, only every five years [46]. It has been noted in national health surveillance system data that estimates of behavioral risk factors obtained from MRP are valid and could be used to characterize local geographic variations in population health indicators when accurate local survey data are not available [26]. Considering the present study results, MRP is a practical approach to reduce estimation bias and produce reliable data interpretations in surveys with limited representative samples.

Regarding the empirical findings, it is relevant to discuss the difference between the inferences, in which the MRP shows significantly higher proportions of compliance with the recommendations of physical activity. Based on our observations, interpretations based on single-level aggregated estimations need to be considered with caution. The potential pitfalls of single-level estimates-based interpretations may be more severe when small sample sizes are available. Considering national health-related surveys, potentially biased inferences may influence the main action’s progress or programs arising from it.

Environmental factors can explain variation in physical activity in leisure-time across state capitals. The Brazilian state capitals present a similarly high Human Development Index [47]. Particularly in Brazil, some government actions are offered to promote physical activity in the health system [48]. Differences between state capitals in a country with enormous social, cultural, economic, and geographic contrasts can be attributed to several factors, such as the heat and humidity, air pollution, cultural differences, public spaces, attitudes toward nature, range of travel modes, and specific cultural contexts [49]. These would represent good examples of potential geographic-level predictors to include within an MRP framework for physical activity when data are available. The importance of geographic predictors in MRP has been noted in other fields [19,21,23]. Our model in this study is an initial example and could be improved by including interactions and other geographic-level predictors [20]. Careful consideration of contextual or geographical variables can improve the prediction precision of MRP [21,23], particularly in predictions for subpopulations [20,21]. Hence, the investigation of more complex MRP models in physical activity research merits further study.

The predicted proportion of physically active individuals in the population across Brazilian state capitals was higher in males up to 40 years. Further, males had higher proportions of physically active individuals than females across all age ranges. These observations are consistent with South American data [5,50] and can be explained in part by biopsychosocial factors [50]. Brown et al. (2016) may explain the results, for instance, as being due to highly uneven access to sports involvement, which predominantly favors males [51]. Another explanation is the physical activity practice in different domains by females, as in the household [5]. However, females’ leisure domain is fewer than for males because of the double journey, particularly in developing countries [50]. For physical activities during leisure (a time reserved for a voluntary choice of activities), combining a feeling of satisfaction, wellbeing, and fun can provide health promotion for physical and mental aspects.

Males and females active in leisure-time physical activity reduced as the age group increased. Leisure is a domain independent of other attributes such as age, exists according to unique opportunities, and is performed voluntarily over short periods with enough recovery time [1]. Therefore, advancing age is attached to making working commitments. Even with more time after the retirement process, routine life reflects the functional capacity and consequences of health outcomes [52]. It follows that older groups had lower levels of physical activity when compared to the youngest ones.

## 5. Conclusions

Our results confirm that MRP is a promising strategy to derive predictions for subpopulations for health-related outcomes and, in particular, physical activity indicators from large surveys. Overall, the MRP is superior to single-level regression estimates, yielding smaller errors and more accurate estimates. Hence, caution in interpreting single-level regression model estimations of physical activity outcomes, particularly with smaller sample size studies, is warranted. Additionally, our models allow for more accurate estimations of target populations. In the present data, younger males from a particular state capital in Brazil were likely to have a minimum amount of time in physical activities for a healthier lifestyle. MRP significantly expands the scope of issues for which researchers can better address participation bias and interpret interactions to estimate descriptive population quantities. The observations present in this study highlight the need for further research, which could, potentially, incorporate more information in the models to better interpret interactions and types of activities across target populations.

## Figures and Tables

**Figure 1 ijerph-18-07477-f001:**
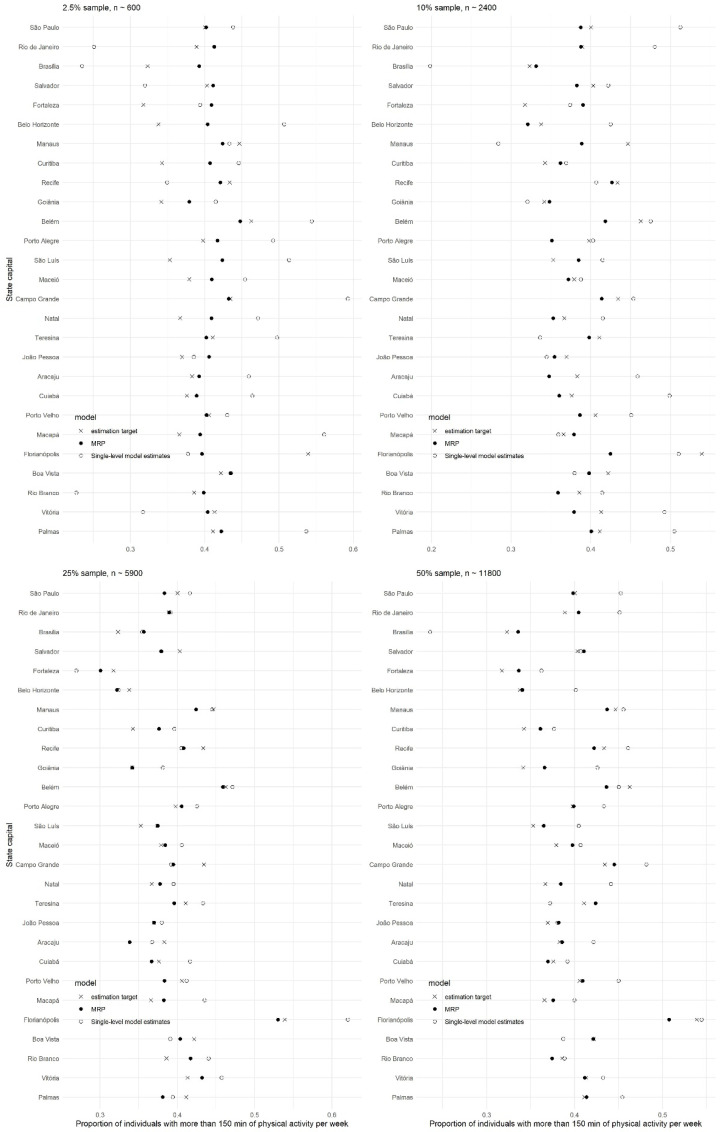
Cross-validation—estimation target (“true value”) against MRP and single-level regression model, and subsample size.

**Figure 2 ijerph-18-07477-f002:**
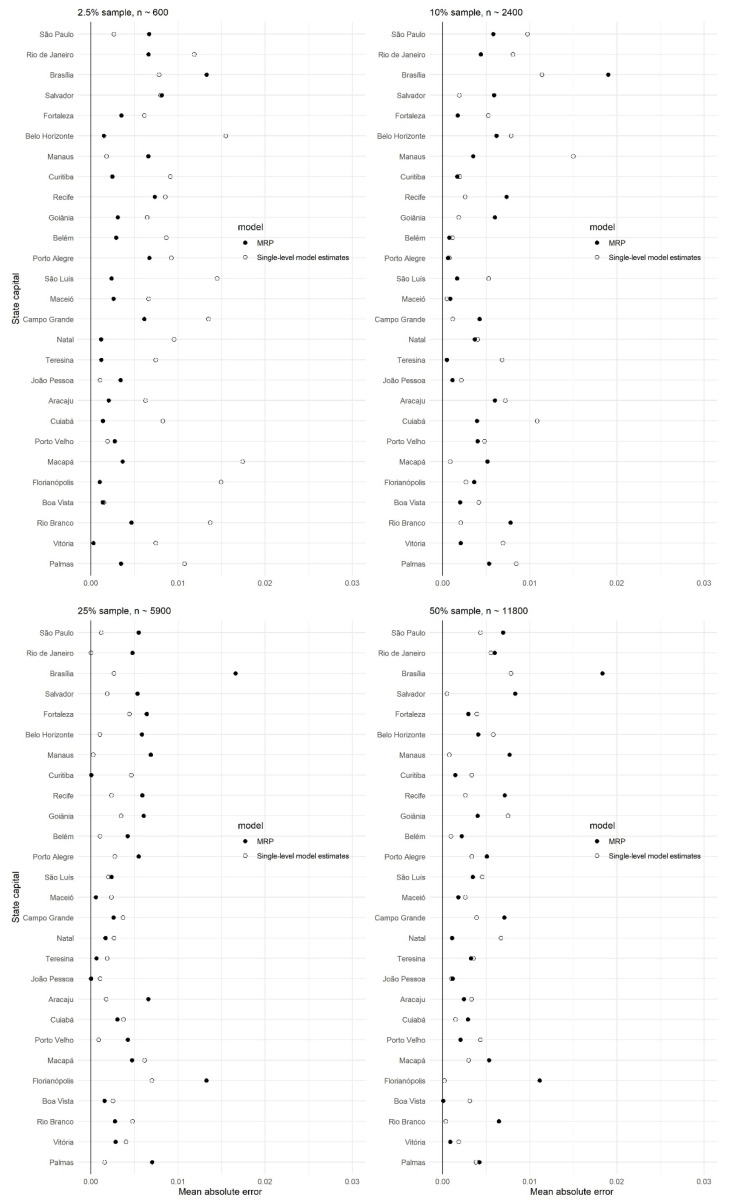
Cross-validation—mean absolute errors by state capital.

**Figure 3 ijerph-18-07477-f003:**
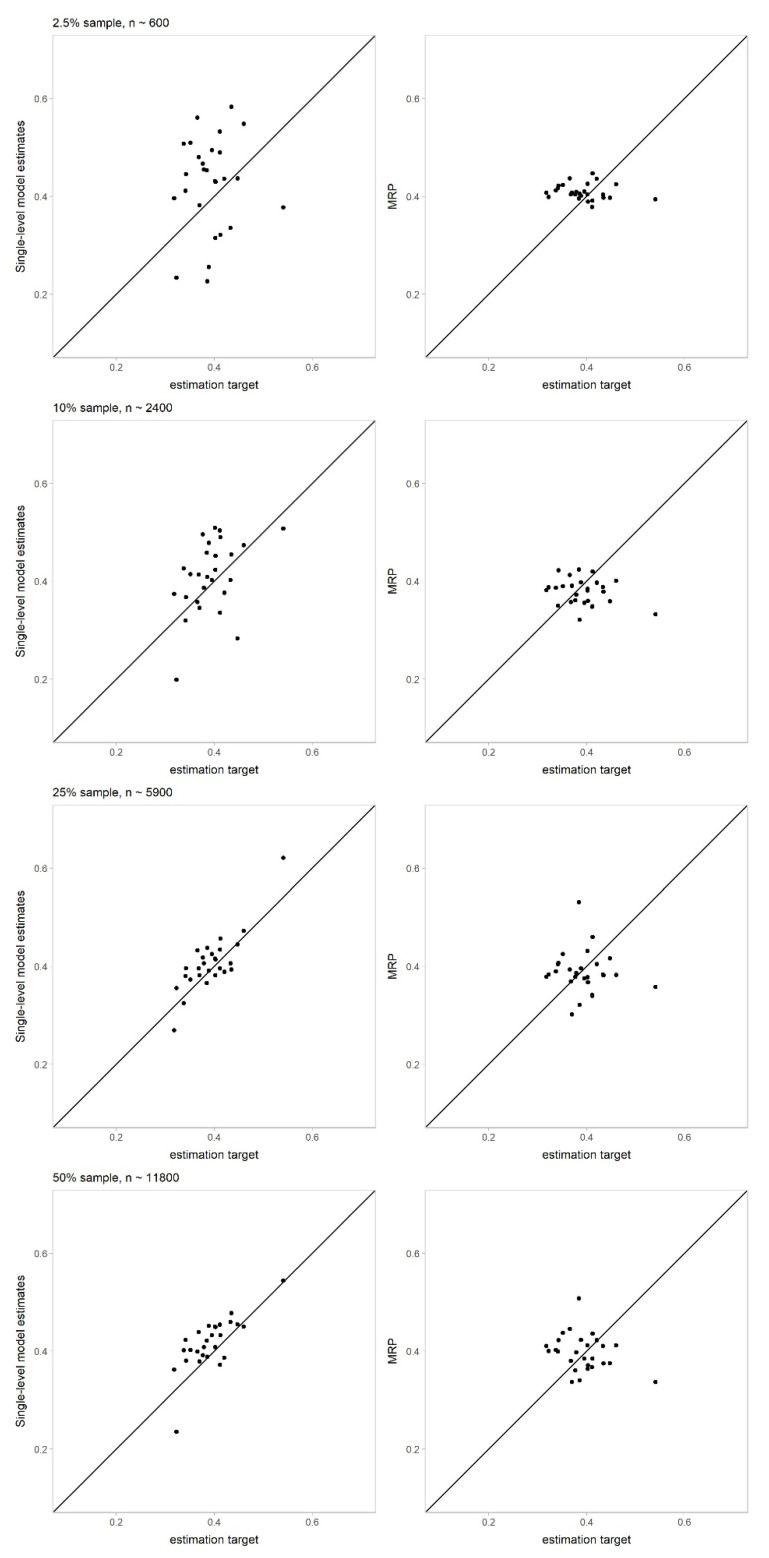
Plotted MRP and single-level regression estimates against estimation target (“true value”) by subsample size.

**Figure 4 ijerph-18-07477-f004:**
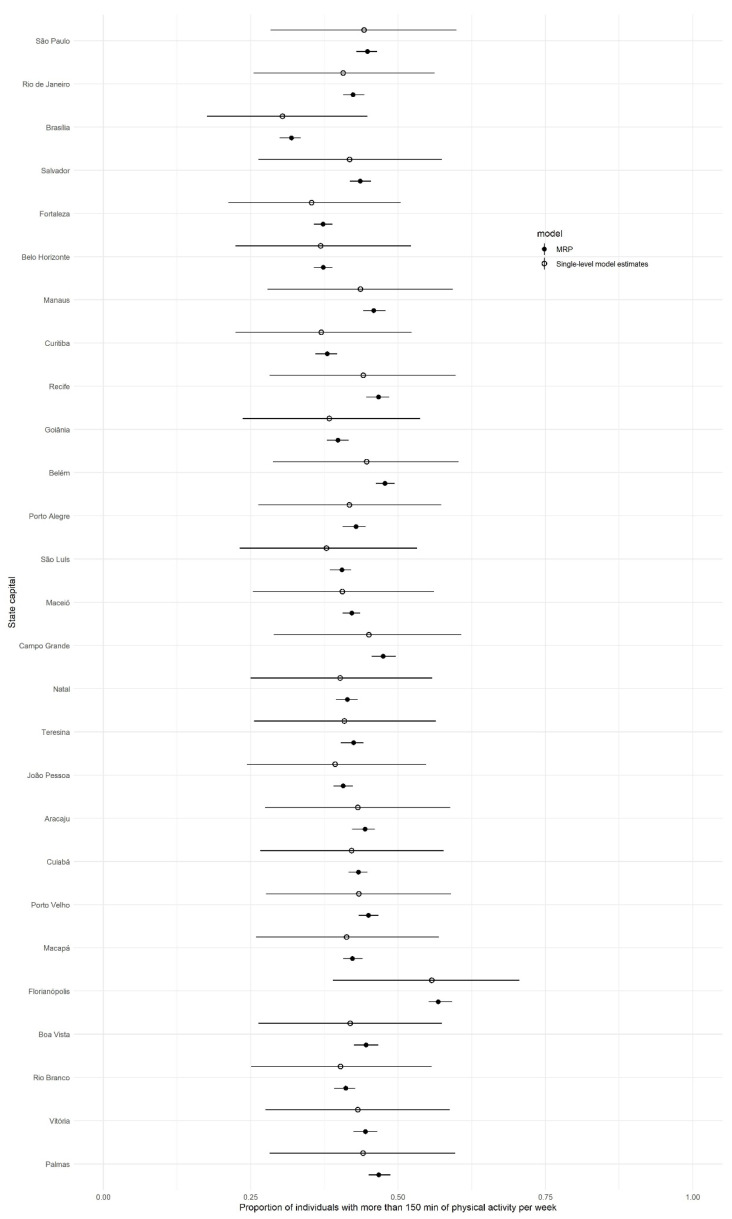
Estimations and 90% credible intervals of the proportion of physically active individuals by state capitals using MRP and single-level regression.

**Figure 5 ijerph-18-07477-f005:**
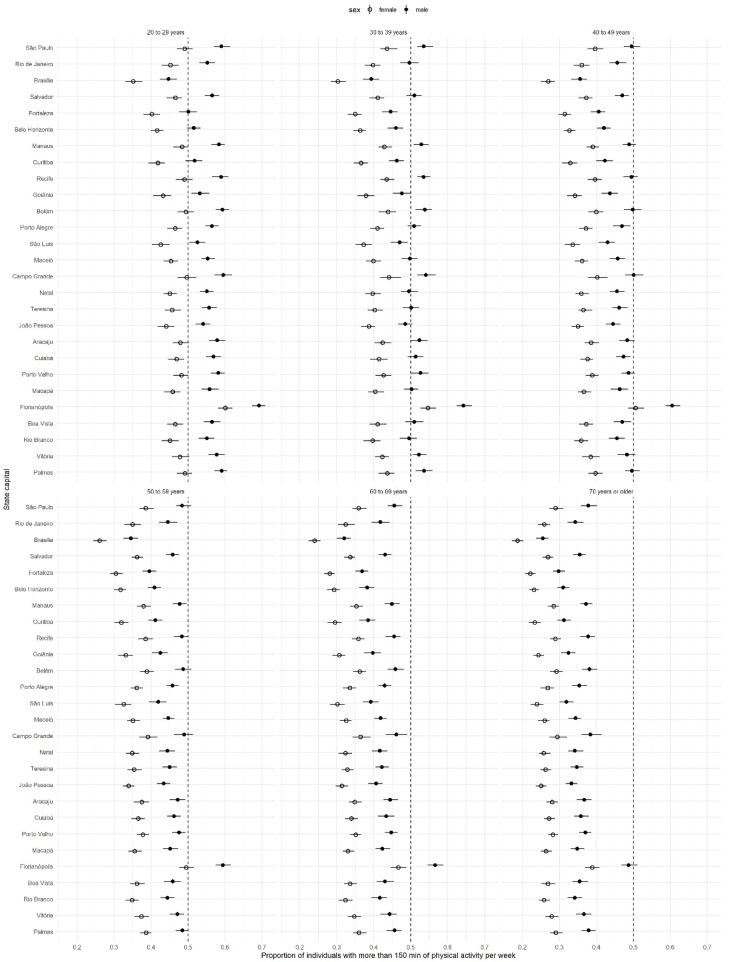
Estimations and 90% credible intervals of the proportion of physically active individuals by gender, age groups, and state capitals based on MRP.

**Table 1 ijerph-18-07477-t001:** Sociodemographic characteristics of the VIGITEL sample.

Variable	Total Sample (%) *n* =51,015	Insufficient Time in PA ^1^ (%) *n* = 31,917	Sufficient Time in PA ^1^ (%) *n* = 19,098
Gender			
Male	15,500 (36.3)	10,243 (32.1)	8257 (43.2)
Female	32,515 (63.7)	21,674 (67.9)	10,841 (56.8)
Age, years			
20 to 29	5999 (11.8)	2989 (9.3)	3010 (15.5)
30 to 39	6710 (13.1)	3773 (11.7)	2937 (15.1)
40 to 49	7767 (15.2)	4680 (15.5)	3087 (15.9)
50 to 59	9974 (19.5)	6140 (19.0)	3834 (19.7)
60 to 69	10,589 (20.8)	6853 (21.3)	3736 (19.2)
≥70	10,673 (20.9)	7807 (24.2)	3529 (14.7)
State			
Acre	1731 (3.4)	1006 (3.2)	725 (3.8)
Alagoas	1996 (3.9)	1294 (4.1)	702 (3.7)
Amapá	1340 (2.6)	771 (2.4)	569 (3.0)
Amazonas	1577 (3.1)	1009 (3.2)	568 (3.0)
Bahia	1969 (3.9)	1347 (4.2)	622 (3.3)
Ceará	1949 (3.8)	1253 (3.9)	696 (3.6)
Distrito Federal	1857 (3.6)	900 (2.8)	957 (5.0)
Espírito Santo	1998 (3.9)	1218 (3.8)	780 (4.1)
Goiás	1981 (3.9)	1264 (4.0)	717 (3.8)
Maranhão	1951 (3.8)	1190 (3.7)	760 (4.0)
Mato Grosso	1970 (3.9)	1210 (3.8)	761 (4.0)
Mato Grosso do Sul	1985 (3.9)	1297 (4.1)	688 (3.6)
Minas Gerais	1934 (3.8)	1209 (3.8)	725 (3.8)
Paraíba	1982 (3.9)	1284 (4.0)	698 (6.7)
Paraná	2022 (4.0)	1286 (4.0)	736 (3.9)
Pará	1986 (3.6)	1334 (4.2)	733 (3.8)
Pernambuco	1983 (3.9)	1334 (4.2)	649 (3.4)
Piauí	1935 (3.8)	1199 (3.8)	736 (3.9)
Rio Grande do Norte	1957 (3.8)	1219 (3.8)	738 (3.9)
Rio Grande do Sul	2025 (4.0)	1370 (4.3)	655 (3.4)
Rio de Janeiro	1981 (3.9)	1363 (4.3)	618 (3.2)
Rondônia	1752 (3.4)	1008 (3.2)	744 (3.9)
Roraima	1557 (3.1)	922 (2.9)	635 (3.3)
Santa Catarina	1918 (3.8)	1200 (3.8)	718 (3.8)
Sergipe	1945 (3.8)	1167 (3.7)	778 (4.1)
São Paulo	1951 (3.8)	1190 (3.7)	534 (2.8)
Tocantins	1941 (3.8)	1401 (4.4)	856 (4.5)

^1^ PA: Physical activity.

## Data Availability

https://osf.io/3qk25/ (accessed on 3 February 2021).

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
