# Peer review of "Using Multilevel Regression and Poststratification to Estimate Physical Activity Levels from Health Surveys"

_ijerph, 2021, doi:10.3390/ijerph18147477_

Round 1

Author Response

Reviewer #1:

  1. The authors should clearly point out the novelty of the proposed approach.

Authors´ reply: We added information in the introduction, mainly presenting the MRP approach. We hope to have addressed the comment appropriately.

  1. It is not well-explained why they used the multilevel logistic regression rather than traditional logistic regression. They need to convince the readers why the latter is inappropriate.

Authors´ reply: Similarly to the first comment, we intended to present the MRP to highlight the advantage of the approach to traditional single level regression approach, which to our knowledge remains the main approach used to deal with extensive data sets with physical activity outcomes.

  1. In Equations (2){(4), how hyperparameters _2 are speci_ed?

Authors´ reply: We added information as suggested. We note that we extended our multilevel model, considering gender as a population-level effects (due to the limited number of groups), but allowed for variation between female and male respondents by age group and state capital. This demanded to extend the specification of the hyperparamenters of the multilevel model.   

  1. How do you interpret Figure 1? Need more elaboration.

Authors´ reply: We changed the figure and report now in figure 1 the estimations of the MRP and single-level model against the estimation target, and in figure two we present the absolute mean error estimations. Here we corrected our code to estimate absolute means, which was not the case in the initial version.

  1. In general, formulas are untidy and not well{aligned.

Authors´ reply: We revised the equations, and followed the journal guideline and the manuscript template as reference.

  1. The term \simple multilevel regression" in lines 12, 80, 215 is a bit confusing, since the word \simple" recalls the simple regression problem.

Authors´ reply: We agree with the reviewer´s point We removed the term.

  1. Rephrase the Abstract. Drop the words such as: background, results, method and conclusions.

Authors´ reply: We followed the journal guidelines and the manuscript template as reference. The terms may be dropped by journal guidelines.

  1. Before moving on to other sections (line 88), by using transitions, mention what will be covered in each section briey.

Authors´ reply: We added the information as suggested.

  1. Line 99, instead of using inactive (none), use (_ 150 min / week)

Authors´ reply: We changed the sentence as suggested.

  1. Eq 6, _0 is not bold.

Authors´ reply: We changed the equation as suggested.

  1. Eq 9, I presume it is 100x 27.

Authors´ reply: We changed the sentence as suggested.

  1. In Table 1, what do the values in (_) represent? How you use them?

Authors´ reply: We thank the reviewer´s observation. We reported the data in brackets as a percentage.

Author Response

Reviewer #2

  1. In the first sentence of the abstract, it is not clear what is meant by the term “units”. Does this refer to individuals or geographic regions? Based on the main text, geographic regions seem likely but this is not clear here.

Authors´ reply: We agree with the reviewer. We changed the sentence as suggested.

  1. The abstract states MRP was evaluated against "single-level aggregated estimates". Similarly, the main text refers to single-level aggregated models. Are these commonly used terms in this field? Perhaps consider using "single-level regression" (which is referred to later in the abstract) or "standard regression models" as these are more commonly used descriptions of the comparison method.

Authors´ reply: We agree with the reviewer´s point. We changed the reference as single-level regression across the text and figures.

  1. The results section of the abstract refers to MRP having smaller mean absolute standard errors and then smaller uncertainty estimates. What is meant by uncertainty estimates? Does this refer to the precision of the estimates?

Authors´ reply: We refer to the uncertainty around the model estimates, in our case to credible intervals.

  1. In the abstract, the objective mentions evaluating MRP in comparison to single-level aggregated estimates while the concluding sentence refers to both single-level aggregated estimates and disaggregation. It’s not clear how disaggregation fits here. Is this a second comparison method?

Authors´ reply: Our aim is to compare MRP and single-level estimates, as the use of the latter remains common with physical activity data. We removed the references to disaggregation, other than its use as target estimate with the simulated samples.

  1. In the abstract, the first sentence of the conclusion does not quite make sense. MRP is an approach for deriving estimates of population descriptive parameters, not for deriving data. What is meant by aggregated-level surveys? This comment also applies to the conclusion section of the main text

Authors´ reply: We simplified the sentence in both abstract and conclusion. Our argument is simply that MRP is valuable to derive estimations of physical activity indicators to target subpopulations from data of large surveys.

  1. The term subsample is used often, and I think inconsistently throughout the manuscript. Does it refer to an individual or a geographic subset of the total sample? For example, it is not clear what this term refers to in the first sentence on page 2. Please check for consistent use of all statistical terms such as population, estimate, uncertainty etc

Authors´ reply: We agree with the reviewer´s point. We removed the term subsample. Only for the cross-validation using different sample sizes we maintained the term.

  1. The initial description of the primary method of interest, MRP, is very brief. A more detailed description in the introduction would be helpful to the reader, for example, explain the use of random effects and the resulting partial pooling that occurs. Why or how is this approach expected to produce (more) reliable estimates of population parameters? Some of this detail is provided later on in the manuscript, but perhaps better placed in the introduction.

Authors´ reply: We agree with the reviewer´s point. We extended the introduction of MRP in the introduction as suggested.

  1. Further to the point above, much of the published literature on MRP highlights the importance of good group- or geographic-level covariates in the multilevel model, particularly when there are a large number of geographic regions in the data, as is the case here with 27 capital cities in Brazil. No geographic-level covariates were included in the multilevel model presented in the manuscript. Were any considered? What covariates might be appropriate for this outcome measure of physical activity? The discussion/conclusion states that further research including consideration of more complex models is required, however, given the existing literature on this, it should perhaps be more directly addressed here.

Authors´ reply: The reviewer´s raises a great point here. There were several reasons to follow a “simpler model”: (i) although the VIGITEL data is extensive and valuable, there are limited matches in the variables for both individual and geographic characteristics with the Brazilian census, limiting the poststratification; (ii) We explored the inclusion of regions at a higher level, which would make sense to account for population dispersion across the Brazilian territory or cultural differences. However, there was not a substantial improvement on what we could learn from this model, and the cost of computation time increased substantially; (iii) we intended to use characteristics that are relatively common in descriptive analysis with physical activity data, and illustrate the potential use of MRP.

However, the comment allowed us to think about our model and consider (in response to a latter comment) the incorporation of gender as population level effect, and the possibility to allow for gender variation across age groups and state capitals. This resulted in a better understanding of the data and the potential interpretations that may arise from the use of MRP.

  1. In many papers published on MRP, binary covariates such as sex have been included in the multilevel model as a fixed effect. Is there any benefit to including sex as a random effect as has been done here?

Authors´ reply: We agree with the point raised. We incorporated gender as population level effect, and the allowed for gender variation across age groups and state capitals. It has been noted that the estimation of variation in groups with less than 3 levels may be problematic. Also it helped with the computation time, which was a limitation with the large dataset.

  1. It would be helpful to provide a brief justification or explanation of the priors chosen in the Bayesian analysis. Were any sensitivity analyses considering alternative priors undertaken?

Authors´ reply: We considered alternative priors, for group-level effects (normal(0,1)), as suggested by the stan development group. Given the computation time, and the large sample size, we did not extend our sensitivity analysis, given the small impact that weakly informative priors would have given the sample size.

  1. The Stan code for the Bayesian model should be provided in the supplementary material.

Authors´ reply: We provide the data sets and code Open Science Foundation repository available at https://osf.io/3qk25/.

  1. On page 4, it states that single-level aggregated models are often used to analyze health related outcomes despite the theoretical and analytical concerns. A couple of references are given here but some detail, even if brief, on what these concerns are would be useful to include.

Authors´ reply: We included the suggestion in the text.

  1. In the MRP approach, the poststratification step is required to generate the estimated proportion of physically active individuals from the multi-level model parameter estimates. Under the standard regression approach, how were the corresponding estimated proportions of physically active individuals obtained from the estimated model coefficients? Was the same poststratification step used?

Authors´ reply: We did not use poststratification for the single-level regression models. We used the posterior predictions of the single-level models, reproducing the inferences commonly made in physical activity studies.

In the fourth paragraph on page 4, there seems to be some formatting issues with suband superscripts, i.e. y_(q,s)^baseline

Authors´ reply: We corrected the descriptions format.

  1. There is an inconsistency on page 4 regarding the number of simulated samples within the split-sample validation; the third paragraph refers to 300 simulations while further down the page refers to 100 simulations. On this page, there is also inconsistent references to four and then five different sample sizes considered.

Authors´ reply: We used 300 simulations, and four sample. We corrected the typos.

  1. For the equations in (9) on page 4, should the s be dropped from the subscripts for both aggregated and MRP methods since these means are calculated across the capital cities? Otherwise, the notation for the mean absolute error terms in (8) and the terms in (9) are identical. Also, in the summation on the right-hand side of these terms in (9) errors summed over s as well as q?

Authors´ reply: We corrected the equation as suggested.

  1. One of the benefits of MRP is that it can be used to obtain reliable estimates of population parameters from highly selected samples. Do you have any information regarding how the VIGITEL sample relates to the target population(s) of interest? Are there likely to be issues of representativeness?

Authors´ reply: We included information about VIGITEL representativeness of the state capitals population.

  1. Figure 1 plots mean absolute error on the x-axis, yet some of the plotted values extend always positive?

Authors´ reply: We thank the reviewer. We had an error in our code and failed to extract absolute mean error values. We correct them to reproduce the equations described for the errors. We now introduce the estimation of MRP and single-level regression against the target prediction, and in figure 2 display the absolute mean errors.

  1. In Figure 1, how are the capital cities ordered on the y-axis? Perhaps a consistent ordering of capital cities across panels would aid comparison. By population size for example. This would help to illustrate how MRP performs relative to the comparison method for capital cities of varying sizes.

Authors´ reply: We agree with the reviewer´s point. We ordered the capitals by population size and standardized it across all figures for an easy reading and comparison.

  1. I’m confused by the first panel of Figure 3. My understanding is that the disaggregated means are being used to represent “true” proportions in the slipt-sample validation. Is this what MRP is being compared to here, what are considered the “true” values? If so, then the focus should be on how close the MRP estimates are to the disaggregated means. Why might MRP be consistently overestimating? If the disaggregated means do represent the true proportions, then it is not clear why they would have uncertainty intervals. Alternatively, if the disaggregated means are being used as an additional comparison method to MRP, this should be introduced earlier. Please clarify what is being depicted here and for what purpose.

Authors´ reply: We agree with the reviewer point. We overstepped here. As noted earlier, we removed all references to disaggregated means, as comparisons between no pooling approach and MRP have been explored in political sciences. Yes, disaggregated means were used to represent “true” proportions in the split-sample validation.

  1. Please clearly state what these intervals in Figure 3 represent. Are they 95% confidence intervals? Or credible intervals? This applies to all figures with intervals.

Authors´ reply: The intervals in figures 4 and 5 represent the 90% credible intervals. We added the information in the figures captions.

  1. Did you consider including interactions between the covariates of age, sex and capital city in your models? Is it reasonable to assume that the associations of the covariates of age and sex with the outcome are the same across all capital cities? Are the same dramatic increases in precision achieved with MRP if the models specify interactions between covariates?

Authors´ reply: We agree with the reviewer´s point. We initially intended to illustrate the use of a “relatively simple” multilevel model. However, even with a limited number of individual and geographical variables, the patterns in the data were clear by allowing variation by gender across age groups and cities. We also merged Figures 4 and 5, as the patterns of variation in physical activity by gender and age groups across state capitals. We believe this was a substantial improve on the manuscript.  

  1. The discussion includes considerable detail that I think would be better presented earlier given these are findings of previously published research into MRP. This includes discussion of the advantages of MRP, use of geographic-level covariates and the ability of MRP to adjust for non-representative samples.

Authors´ reply: We thank the reviewer´s comment. We adjusted the text as suggested.

  1. The manuscript referred to supplementary material, but I was unable to view this as part of the review.

Authors´ reply: The supplementary material is available in the OSF repository, at https://osf.io/3qk25/.

Round 2

Reviewer 2 Report

Using Multilevel Regression and Poststratification to Estimate Physical Activity Levels from Health Surveys

  • In the abstract, you refer to “various subsample proportions tested”. This refers to the cross validation doesn’t it? I would make this clear here, i.e. “We used various approaches to compare the MRP and single-level model (complete-pooling) estimates, including cross-validation with various subsample proportions tested.”
  • In the introduction, the word “subsample” from the previous draft seems to have been universally replaced with “sub-population”. In most cases, this is fine but not all. For example, on the third line of page 2, I think the most appropriate word is sample. Please again check for consistent and correct use of all statistical terms such as population, sample, estimate, uncertainty etc.
  • The revised model specification has become a lot more complicated. I’m not quite sure why the effects of age and state capital have been changed from random effects to fixed effects. Under this specification, there can be no partial pooling across these categories. I think the model in the original submission is preferrable, but with sex as a fixed effect (since it is a binary indicator variable).
  • I would also remove the prior distributions from the set of equations when you define the multilevel model. You have specified them at the beginning of page 5 which is enough.
  • For the single-level regression model specified in equation 15, shouldn’t age group and state capital now be fixed effects as they are categorical variables with more than two levels? In its current form it looks as though you are fitting a single linear regression coefficient for each of these variables.
  • I’m still not sure the subscripts are correct for the summary measures in equations 17 and 18. What you are summing over should not appear in the subscript on the left-hand side of each equation.
  • In the caption of Figure 1 (and the text reference), refer to the cross-validation subsample size rather than just sample size as this is clearer.
  • When you say “Particularly for small sample sizes, the mean absolute errors were consistently smaller for the MRP estimates”, here you are referring to state capital sample sizes so make this clear too in order to differentiate from the cross-validation subsample size comparisons earlier in this paragraph.
  • Looking at Figure 2, mean absolute errors for MRP were not always lower than for simple regression, so rather than saying “the mean absolute errors were consistently smaller for the MRP estimates”, it might be more appropriate to say “the mean absolute errors were more often smaller for MRP estimates”.
  • The scatterplots in Figure 3 for MRP do not look as good as the corresponding scatterplots for MRP in the previous version of the manuscript. I wonder whether this is a result of the different model specification, i.e. age and state capital as fixed effects. Why do the scatterplots for the simple regression model also look different, particularly for small cross-validation subsample sizes? Isn’t this the same model?
  • Same comment as above regarding the caption and references to Figure 3, instead of sample size, I suggest referring to cross-validation subsample size. Similarly, in the discussion, whenever you refer to sample size, clarify whether it is cross-validation subsample size, state capital sample size or just more generally, study sample size.
  • Figure 4 shows a dramatic increase in precision using MRP compared to simple regression. This can be attributed to partial pooling in the multilevel model and also because you have used a very simple model. I think it is worth addressing this in the discussion. A more complex model including interactions between covariates may reduce the level of partial pooling and therefore the precision of subgroup estimates.
  • In the discussion you state, “Differences between state capitals in a country with enormous social, cultural, economic, and geographic contrasts Brazil can be attributed to several factors such as the heat and humidity, air pollution, cultural differences, public spaces, attitudes toward nature, range of travel modes, and specific cultural contexts [49].” These would all be good examples of potential geographic-level covariates to include in the multilevel model for physical activity if data were available. Perhaps you could make this point more explicitly when you suggest further investigation of more complex models.

Author Response

Reviewer #2

Dear Authors

Overall, there has been an improvement in the quality of the manuscript particularly regarding the presentation of results. However, there are some new major issues in this revised version, particularly regarding changes made to the statistical models fitted. I would like to encourage the authors to resubmit the manuscript since it fits perfectly in our special issue.

In the abstract, you refer to “various subsample proportions tested”. This refers to the cross validation doesn’t it? I would make this clear here, i.e. “We used various approaches to compare the MRP and single-level model (complete-pooling) estimates, including cross-validation with various subsample proportions tested.”

Authors´ reply: We changed the sentence in the abstract as suggested.

In the introduction, the word “subsample” from the previous draft seems to have been universally replaced with “sub-population”. In most cases, this is fine but not all. For example, on the third line of page 2, I think the most appropriate word is sample. Please again check for consistent and correct use of all statistical terms such as population, sample, estimate, uncertainty etc.

Authors´ reply: We revised the use of sample or  sub-population. Hopefuly the text now uses correct terms.

The revised model specification has become a lot more complicated. I’m not quite sure why the effects of age and state capital have been changed from random effects to fixed effects. Under this specification, there can be no partial pooling across these categories. I think the model in the original submission is preferrable, but with sex as a fixed effect (since it is a binary indicator variable). I would also remove the prior distributions from the set of equations when you define the multilevel model. You have specified them at the beginning of page 5 which is enough.

Authors´ reply: We modeled gender fixed effect allowing to vary randomly by age group and state capital. The expression of our model was tricky. Likely a simpler notation removing the prior specifications may suffise. However, we agree with the reviewer, and a simpler model addresses our concerns for this manuscript. We adopted the model suggested, which is almost identical to our initial model.

We also followed the suggestion and removed the specification of the priors, as it is redundant with the text.

For the single-level regression model specified in equation 15, shouldn’t age group and state capital now be fixed effects as they are categorical variables with more than two levels? In its current form it looks as though you are fitting a single linear regression coefficient for each of these variables.

Authors´ reply: We added information about the variables in the model as well as the corresponding subscripts. We hope the notation and information reflects our models correctly.

I’m still not sure the subscripts are correct for the summary measures in equations 17 and 18. What you are summing over should not appear in the subscript on the left-hand side of each equation.

Authors´ reply: We corrected the subscripts of the equations. We follow the notation and reference of Lax and Phillips (2009, doi:10.1111/j.1540-5907.2008.00360.x.).

In the caption of Figure 1 (and the text reference), refer to the cross-validation subsample size rather than just sample size as this is clearer.

Authors´ reply: We corrected the captions and across the text referring to the subsamples of the cross-validation.

When you say “Particularly for small sample sizes, the mean absolute errors were consistently smaller for the MRP estimates”, here you are referring to state capital sample sizes so make this clear too in order to differentiate from the cross-validation subsample size comparisons earlier in this paragraph. Looking at Figure 2, mean absolute errors for MRP were not always lower than for simple regression, so rather than saying “the mean absolute errors were consistently smaller for the MRP estimates”, it might be more appropriate to say “the mean absolute errors were more often smaller for MRP estimates”.

Authors´ reply: We thank the reviewer suggestion. We adjusted the interpretation as suggested.

The scatterplots in Figure 3 for MRP do not look as good as the corresponding scatterplots for MRP in the previous version of the manuscript. I wonder whether this is a result of the different model specification, i.e. age and state capital as fixed effects. Why do the scatterplots for the simple regression model also look different, particularly for small cross-validation subsample sizes? Isn’t this the same model?

Same comment as above regarding the caption and references to Figure 3, instead of sample size, I suggest referring to cross-validation subsample size. Similarly, in the discussion, whenever you refer to sample size, clarify whether it is cross-validation subsample size, state capital sample size or just more generally, study sample size.

Authors´ reply: Here we are having some trouble to understand whats going on with our code and models. Despite using the code (although we have made some minor corrections in some errors found in the code) and the same data, we cannot replicate the initial plots reported in figure 3 for MRP predictions. A possible cause may be not setting a seed function for result replication initially. This may explain why the plot for single-level model predictions look slightly different.  However, this does not explain the behavior of the preditions of the MRP preditions, which seems to be similar, irrespective of of model complexity. Maybe an independent inspection of our code and data available in the OSF repository may help.

Figure 4 shows a dramatic increase in precision using MRP compared to simple regression. This can be attributed to partial pooling in the multilevel model and also because you have used a very simple model. I think it is worth addressing this in the discussion. A more complex model including interactions between covariates may reduce the level of partial pooling and therefore the precision of subgroup estimates.

In the discussion you state, “Differences between state capitals in a country with enormous social, cultural, economic, and geographic contrasts Brazil can be attributed to several factors such as the heat and humidity, air pollution, cultural differences, public spaces, attitudes toward nature, range of travel modes, and specific cultural contexts [49].” These would all be good examples of potential geographic-level covariates to include in the multilevel model for physical activity if data were available. Perhaps you could make this point more explicitly when you suggest further investigation of more complex models.

Authors´ reply: We thank the reviewer´s suggestion. We included the argument in the discussion.